# SILAC-Based Quantitative Proteomic Analysis of Oxaliplatin-Resistant Pancreatic Cancer Cells

**DOI:** 10.3390/cancers13040724

**Published:** 2021-02-10

**Authors:** Young Eun Kim, Eun-Kyung Kim, Min-Jeong Song, Tae-Young Kim, Ho Hee Jang, Dukjin Kang

**Affiliations:** 1Center for Bioanalysis, Division of Chemical and Medical Metrology, Korea Research Institute of Standards and Science, Daejeon 34113, Korea; yekim@kriss.re.kr; 2School of Earth Sciences and Environmental Engineering, Gwangju Institute of Science and Technology, Gwangju 61005, Korea; kimtaeyoung@gist.ac.kr; 3Department of Biochemistry, College of Medicine, Gachon University, Incheon 21999, Korea; ekkim@gachon.ac.kr (E.-K.K.); neptune6nrg@hanmail.net (M.-J.S.)

**Keywords:** quantitative proteomics, SILAC, pancreatic cancer, drug resistance, oxaliplatin

## Abstract

**Simple Summary:**

Resistance to oxaliplatin remains a major challenge in pancreatic cancer therapy. However, molecular mechanisms underlying oxaliplatin resistance in pancreatic cancer is still unclear. The aim of this study was to identify global changes of proteins involved in oxaliplatin resistance in pancreatic cancer cells, thereby elucidating the multiple mechanisms of oxaliplatin resistance in pancreatic cancer. We presented the quantitative proteomic profiling of oxaliplatin-resistant pancreatic cancer cells via a stable isotope labelling by amino acids in cell culture (SILAC)-based shotgun proteomic approach. Multiple biological processes including DNA repair, cell cycle process, and type I interferon signaling pathway were enriched in oxaliplatin-resistant pancreatic cancer cells. Furthermore, we demonstrated that both Wntless homolog protein (WLS) and myristoylated alanine-rich C-kinase substrate (MARCKS) could participate in oxaliplatin resistance in pancreatic cancer cells.

**Abstract:**

Oxaliplatin is a commonly used chemotherapeutic drug for the treatment of pancreatic cancer. Understanding the cellular mechanisms of oxaliplatin resistance is important for developing new strategies to overcome drug resistance in pancreatic cancer. In this study, we performed a stable isotope labelling by amino acids in cell culture (SILAC)-based quantitative proteomics analysis of oxaliplatin-resistant and sensitive pancreatic cancer PANC-1 cells. We identified 107 proteins whose expression levels changed (thresholds of 2-fold changes and *p*-value ≤ 0.05) between oxaliplatin-resistant and sensitive cells, which were involved in multiple biological processes, including DNA repair, cell cycle process, and type I interferon signaling pathway. Notably, myristoylated alanine-rich C-kinase substrate (MARCKS) and Wntless homolog protein (WLS) were upregulated in oxaliplatin-resistant cells compared to sensitive cells, as confirmed by qRT-PCR and Western blot analysis. We further demonstrated the activation of AKT and β-catenin signaling (downstream targets of MARCKS and WLS, respectively) in oxaliplatin-resistant PANC-1 cells. Additionally, we show that the siRNA-mediated suppression of both MARCKS and WLS enhanced oxaliplatin sensitivity in oxaliplatin-resistant PANC-1 cells. Taken together, our results provide insights into multiple mechanisms of oxaliplatin resistance in pancreatic cancer cells and reveal that MARCKS and WLS might be involved in the oxaliplatin resistance.

## 1. Introduction

Pancreatic cancer is one of the most lethal cancers, with the five-year survival rate of 8%, the lowest survival rate among other common types of cancer [1]. Despite recent advances in cancer therapeutics, pancreatic cancer still has a poor prognosis because it has no distinctive symptoms in early stages, and therefore often spread to other organs before it is diagnosed. In addition, both intrinsic and acquired chemoresistance are a main cause of failure in chemotherapy treatment of pancreatic cancer [2,3,4].

Oxaliplatin is a platinum-based chemotherapy drug used in the treatment of various types of cancers, including pancreatic, colorectal, and gastric cancers [5,6,7]. The combination of oxaliplatin with other chemotherapy drugs (5-FU, leucovorin, and irinotecan) is one of the standard regimens in first-line treatment for pancreatic cancer [8]. Similar to other platinum drugs, oxaliplatin is known to cause DNA damage by the formation of platinum-DNA adducts, resulting in cell toxicity and death [9,10]. Although the use of oxaliplatin is effective in the treatment of cancers, acquired resistance to oxaliplatin often occurs in patients, which leads to therapeutic failures. Many studies have reported several different mechanisms of resistance to oxaliplatin in the acquired oxaliplatin-resistant cancer cell lines [10,11,12,13], which include the regulation of cellular transport and detoxification [11], the enhancement of DNA repair system [13], and the activation of NF-kB signaling [12]. However, understanding of multiple mechanisms for acquired oxaliplatin resistance remains a challenge in pancreatic cancer treatments.

Mass spectrometry-based proteomics has become a powerful tool to explore multiple mechanisms of chemoresistance in cancer cells, which allows the global identification and quantification of proteins associated with drug resistance [14,15,16]. For example, an earlier study has reported the comparative proteomic profiling between oxaliplatin sensitive and resistant human colorectal cancer cells [16]. These authors detected down-regulation of pyruvate kinase M2 (PK-M2) in oxaliplatin-resistant cells and further demonstrated an inverse relationship between PK-M2 expression and oxaliplatin resistance in patients with colorectal cancer.

The aim of this study is to investigate the global proteomic changes associated with acquired oxaliplatin resistance in pancreatic cancer cells. We established oxaliplatin-resistant PANC-1 cells by stepwise exposure to increasing concentration of oxaliplatin. A stable isotope labelling by amino acids in cell culture (SILAC)-based quantitative proteomics analysis of oxaliplatin sensitive and resistant PANC-1 (PANC-1R) cells was performed using two-dimensional nanoflow liquid chromatography-tandem mass spectrometry (2D-nLC-MS/MS). A number of proteins involved in DNA repair, cell cycle process, and type I interferon signaling pathway were significantly changed in PANC-1R cells compared to sensitive cells. Moreover, we identified myristoylated alanine-rich C-kinase substrate (MARCKS) and Wntless homolog protein (WLS) as highly upregulated proteins in PANC-1R cells, and validated these using qRT-PCR and Western blotting. Finally, we then explored the roles of MARCKS and WLS in oxaliplatin resistance using siRNA silencing. 

## 2. Results

### 2.1. The Establishment and Validation of Oxaliplatin-Resistant PANC-1 Cells

The human pancreatic cell line PANC-1 was subjected to gradually increasing concentrations of anticancer-drug. To examine the acquired drug resistance of PANC-1 cells, drug sensitivity to oxaliplatin was measured in parental and drug-resistant cells using a cell viability assay. The cell viability of parental PANC-1 cells was decreased depending on the concentration of oxaliplatin, whereas the oxaliplatin-resistant PANC-1 (PANC-1R) cells showed a high cell survival rate, even at high concentrations of oxaliplatin (Figure 1A). To examine the potential of tumorigenesis in oxaliplatin-resistant PANC-1R cells, we performed colony formation assay. The colony-forming ability of PANC-1R cells was increased relative to the parental PANC-1 cells (Figure 1B). These results indicate that PANC-1R cells exhibit the acquired chemoresistant features for oxaliplatin.

### 2.2. Quantitative Proteomic Analysis of Oxaliplatin-Resistant and Sensitive PANC-1 Cells

To study changes in protein expression associated with oxaliplatin resistance in PANC-1 cells, SILAC-based quantitative proteomic analysis was performed using online 2D-nLC-MS/MS. To this end, PANC-1 cells were metabolically labelled with two “heavy” isotope amino acids (^13^C_6_-Arg and ^15^N_2_^13^C_6_-Lys), while PANC-1R cells were cultured with their “light” amino acid counterparts (^12^C_6_-Arg and ^14^N_2_^12^C_6_-Lys) (Figure 2A). Equal amounts of PANC-1R (light) and PANC-1 (heavy) cell lysates were combined, followed by tryptic digestion and online 2D-nLC-MS/MS analysis. Quality assessments of the proteomic dataset are shown in Figure 2B and Appendix A. There are linear correlations between biological replicates with R squared values ranged from 0.797 to 0.877 (Figure 2B), indicating good reproducibility. Histograms of normalized log_2_ (PANC-1R/PANC-1) were normally distributed (Appendix A). 

A total of 3544 proteins were commonly quantified in both PANC-1 and PANC-1R cells, considering only proteins that were quantified in at least three of the six replicates (Appendix A). Among these, 107 proteins were significantly changed between PANC-1 and PANC-1R cells with thresholds of 2-fold changes and *p*-value ≤ 0.05 (Figure 2C). Compared with oxaliplatin sensitive PANC-1 cells, 54 proteins were upregulated, and 53 proteins were downregulated in PANC-1R cells (Appendix A). To gain more insight into the biological functions of significantly changed proteins, GO enrichment analysis was performed using DAVID. All enriched GO terms, including biological processes and molecular functions, are shown in Appendix A. We represented the top 3 ranked GO terms of biological process according to the statistical significance of the enrichment (Figure 2D). Noticeably, base-excision repair was enriched in upregulated proteins. The base-excision repair pathway is a critical DNA repair system to correct damaged bases from oxidation, alkylation, and deamination [17]. Since DNA repair system is required to overcome platinum drug-induced DNA damages, up-regulation of base-excision repair pathway leads to oxaliplatin resistance [18,19]. Out of the GO terms enriched in downregulated proteins, type I interferon signaling pathway plays an important role in anti-cancer immunity, which promotes inhibition of cancer cells and anti-cancer immune responses [20]. Moreover, a previous study has reported that the down-regulation of ISG15 (type I interferon signaling protein) induced the resistance to cisplatin (one of the platinum drugs) in colorectal cancer cells [21], which is consistent with our proteomic result. Reactome pathway analysis also revealed that downregulated proteins were significantly enriched in interferon signaling, interferon alpha/beta signaling, and DDX58/IFIH1-mediated induction of interferon alpha/beta signaling (Figure 3). However, there was no significant enrichment of reactome pathway for upregulated proteins. 

Next, we constructed the PPI networks for up/downregulated proteins in PANC-1R cells using STRING database and mapped with Cytoscape (Figure 4A,B). Further PPI network analysis was performed by MCODE cluster and GO term analysis. In the PPI network of upregulated proteins, the CCNB1-NCAPH-KIF2C-ANLN module (MCODE score = 4) was identified that are involved in cell cycle progress. Previous studies have been reported that oxaliplatin induced the G2/M arrest of the cell cycle, resulted in cancer cell growth inhibition [22,23,24]. The abrogation of G2/M arrest is known to be one of the mechanisms of platinum drug resistance [25], which is consistent with our proteomic result that proteins involved in cell cycle process were upregulated in PANC-1R cells. In addition, base excision repair-related proteins including TP53, LIG1, and HMGA2 were also indicated in the PPI network, although the module of these proteins was not observed by cluster analysis. In the PPI network of downregulated proteins, the most significant module (MCODE score = 9) consisted of 9 nodes (IFIT1, IFIT2, IFIT3, IFIH1, ISG15, OASL, DDX58, DDX60, and HERC5) with 36 edges, which is functionally associated with type I interferon signaling pathway.

### 2.3. Verification of Differentially Expressed Proteins between Oxaliplatin Sensitive and Resistant PANC-1 Cells by Western Blot

To verify quantitative proteomics datasets, Western blotting was performed for six significantly changed proteins that are cellular tumor antigen p53 (p53), G2/mitotic-specific cyclin-B1 (Cyclin B1), superoxide dismutase (SOD2), interferon-induced protein with tetratricopeptide repeats 3 (IFIT3), ubiquitin-like protein ISG15 (ISG15), and heme oxygenase 1 (HO-1). Among these, p53 and Cyclin B1 were upregulated proteins in PANC-1R cells, which are functionally involved in base-excision repair and cell cycle process, respectively. In addition, IFIT3, ISG15, SOD2, and HO-1 were downregulated protein in PANC-1R cells, of which IFIT3 and ISG15 belong to the type I interferon signaling pathway. Resultingly, the changes in expression levels of these six proteins were consistent with their quantitative proteomic results (Figure 5A,B).

### 2.4. MARCKS or WLS Was a Significant Factor for Chemoresistant in PANC-1R Cells

Based on quantitative proteomic analyses, the proteins implicated in base-excision repair and cell cycle progress were upregulated in PANC-1R cells, which are well-known mechanisms of contributing to oxaliplatin resistance platinum drug resistance [18,19,25,26]. Therefore, we explored other potential targets to elucidate additional mechanisms of oxaliplatin resistance in PANC-1 cells. Out of top 10 most highly expressed proteins (Appendix A), we validated the functional roles of MARCKS and WLS in PANC-1R cells. MARCKS is a substrate of protein kinase C (PKC), and it has previously reported that inhibition of MARCKS overcome the drug resistance in multiple myeloma cells [26]. WLS—regulates the sorting and secretion of Wnt proteins [27]—plays a key role in the activation of Wnt signaling, which is also known to confer drug resistance in cancer therapy [28]. 

MARCKS was highly expressed in PANC-1R cells (Appendix A). MARCKS is involved in transducing receptor-mediated signals into intracellular kinases, such as Akt and PKC [29,30,31]. The SILAC ratio of MARCKS protein expression level was 6-fold higher in PANC-1R cells compared to PANC-1 cells (Figure 6A). The mRNA level of MARCKS measured by qRT-PCR was also 6-fold higher in PANC-1R cells compared to PANC-1 cells (Figure 6B). To confirm the protein level and activity of MARCKS, we examined the levels of MARCKS and its downstream protein using Western blot analysis. We found that increased phosphorylated level and total protein level of MARCKS induced AKT phosphorylation in PANC-1R cells (Figure 6C). 

Wntless homolog protein (WLS, Evi, or GPR177) was also detected to be highly expressed in PANC-1R cells (Appendix A). The SILAC ratio of WLS protein expression level was 4-fold higher in PANC-1R cells compared to PANC-1 cells (Figure 6D). The mRNA level of WLS is also elevated in PANC-1R cells (Figure 6E). WLS is essential for β-catenin signaling [32,33]. To check the activity of WLS, we examined the level of β-catenin and its target cyclin D1 [34,35]. Up-regulation of WLS in PANC-1R cells increased the expression of β-catenin and cyclin D1 (Figure 6F). 

### 2.5. Inhibition of MARCKS and WLS Increased Oxaliplatin-Mediated Cell Death in Chemoresistant PANC-1R Cells

Next, we explored whether down-regulation of MARCKS and WLS in PANC-1R cells affects cell survival for oxaliplatin treatment. When silencing in PANC-1R cells using siRNA specific for MARCKS, cell viability to oxaliplatin was 10% lower at 20 μg/mL of oxaliplatin and 14% lower at 50 μg/mL of oxaliplatin compared to control cells (siCon) (Figure 7A,B and Appendix A). The cell viability of the knockdown of WLS was 16% lower at 20 μg/mL of oxaliplatin and 20% lower at 50 μg/mL of oxaliplatin than control (Figure 7A,B, and Appendix A). We examined the effect of the double knock downed MARCKS and WLS in cell viability to oxaliplatin (Figure 7C,D). Cellular viability with 20 μg/mL of oxaliplatin was 30% lower in the double knock down group. The treatment of 50 μg/mL of oxaliplatin in the double knock down had 40% lower of viability compared to siCON. These results indicated that drug resistance in PANC-1R cells was regulated by the association of several factors rather than by a single factor.

## 3. Discussion

To understand the mechanism of oxaliplatin-resistant in pancreatic cancer cells, we successfully established oxaliplatin-resistant pancreatic cancer PNAC-1 cell lines by a stepwise increase of oxaliplatin concentration in a culture medium. Using SILAC-based 2D-nLC-MS/MS, the quantitative proteomic analysis was performed across PANC-1R and PANC-1 cells. We identified a number of significantly changed proteins in oxaliplatin-resistant cells compared with sensitive cells, which were associated with multiple biological processes, including base-excision repair, cell cycle process, and type I interferon signaling pathway.

We identified the up-regulation of base-excision repair in PANC-1R cells compared to PANC-1 cells (Figure 2D and Appendix A). Base-excision repair is one of the major DNA repair systems for oxidative DNA damages, which is a known pathway involved in resistance to oxaliplatin [13,36]. Because oxaliplatin induces the formation of free radicals as well as oxaliplatin-DNA adducts, exposure to oxaliplatin causes oxidative DNA damages and subsequently cytotoxicity [11,37]. Therefore, an increase of base-excision repair capacity could contribute the resistance to oxaliplatin-induced cytotoxicity. 

Wild-type p53 is a tumor suppressor that regulates cell death, apoptosis, senescence, and cell cycle when activated by oncogenic signal and DNA damages [38]. However, p53 mutants promote the oncogenic process to activate tumor development and chemoresistance in several cancers [39,40,41,42,43,44]. PANC-1 cells have mutant p53 [45]. In PANC-1 cells, mutant p53 induces the expression of cyclin B1 by gemcitabine and enhances cell survival against anti-cancer drugs [44]. The proteomic analysis and western blotting also showed more high level of p53 in PANC-1R than PANC-1 (Figure 5B).

The over-activation of cell cycle progression is one of the chemoresistance mechanisms in various cancers and induces the change of cellular metabolism [46,47]. High levels of cyclin B1 and D1 were observed in oxaliplatin-resistant colon cancer cells [48]. Our results also showed that the level of cyclin B1 was higher in PANC-1R than PANC-1 (Figure 5B). These data indicate the possibility that chemoresistance can be modulated through common factors.

Our study also identified type I interferon signaling-related proteins (IFIT1, IFIT2, IFIT3, OASL, and ISG15) that were downregulated in PANC-1R cells (Figure 2D and Figure 3), and further confirmed the expression level of IFIT3 and ISG15 by Western blot (Figure 5B). Previous studies reported the role of type I interferon signaling in resistance to platinum drug [21,49,50,51]. Huo et al. reported that silencing of ISG15 increased cisplatin resistance in colorectal cancer A549 cells by the increase of p53 stability [21], which is consistent with our findings of a down-regulation of ISG15 and up-regulation of p53 in PANC-1R cells. In addition, exogeneous type I interferon increased the sensitivity to cisplatin in cancer cells [51]. In contrast, another study has shown that the activation of the STAT1 pathway and downstream interferon-stimulated genes contributes to platinum drug resistance in human ovarian cancer cells [49]. Although this study was focused on upregulate proteins in PANC-1R cells, further investigation on the role of type I interferon signaling in oxaliplatin resistant pancreatic cancer cells will provide better understanding of chemoresistance in pancreatic cancer. 

It is notable that the expression of MARCKS was upregulated at both the mRNA and protein levels in PANC-1R cells (Figure 6A–C). MARCKS is a substrate of protein kinase C that plays a regulatory role in various cellular functions, such as actin cytoskeleton, cell migration, and cell cycles [31], which had not been previously identified to be involved in oxaliplatin resistance. Recent studies have shown that MARCKS regulates intracellular phosphatidylinositol 4, 5-bisphosphate (PIP2) levels and thereby activating PI3K/AKT signaling [52,53,54]. In addition, MARCKS knockdown reduces phosphorylation of PI3K and AKT in non-small-cell lung cancer (NSCLC) cells and renal cell carcinoma (RCC) [29]. In the present study, we show an increase in the levels of AKT phosphorylation (Ser473 and Thr308) in PANC-1R cells. (Figure 6C). Since activation of the PI3K/AKT signaling pathway contributes to oxaliplatin resistance in hepatocellular carcinoma [55], colon cancer [56], and cholangiocarcinoma cells [57], it is possible that oxaliplatin resistance was acquired by activation of MARCKS and its downstream AKT signaling in pancreatic cancer cells. 

WLS is a transmembrane protein that regulates tracking and secretion of Wnt signaling molecules [58]. Secreted Wnt ligands bind to Frizzled receptors and LRP 5/6 co-receptors, resulting in the activation of Wnt/β-catenin signaling pathway [58,59]. Wnt/β-catenin signaling plays an important role in the cellular and developmental process and is aberrantly activated in various types of cancer [59,60,61]. Several previous studies demonstrated the association of the Wnt/β-catenin pathway with chemoresistance in cancer cells [62,63,64]. Kukcinaviciute et al. have reported the up-regulation of the Wnt pathway in oxaliplatin resistance colorectal cancer cells HCT116 [62]. Our proteomic results have shown the up-regulation of WLS in PANC-1 R cells, and it was confirmed by qRT-PCR and Western blot (Figure 6D–F). We also observed the overexpression of β-catenin and its target gene cyclin D1 in PANC-1R cells by Western blot (Figure 6F), which indicates the activation of the Wnt/β-catenin pathway in oxaliplatin-resistant cells, compared to sensitive cells. These results suggested that activation of Wnt/β-catenin signaling might lead to oxaliplatin resistance in pancreatic cancer cells. 

Wnt affects MARCKS activation, and MARCKS also contributes to Wnt-mediated invasion [65]. Although the proteomic analysis does not directly detect the Wnt protein, high levels of WLS may be causative of inducing the secretion of Wnt, that promotes the activation of MARCKS and its downstream pathways in oxaliplatin-resistance pancreatic cancer. Therefore, we investigated the signal crosstalk between WLS and MARCKS in oxaliplatin resistance and demonstrated the sensitivity to oxaliplatin through knockdown assay of single gene or double genes. Dual suppression of MARCKS and WLS showed an additive effect on increasing oxaliplatin sensitivity of PANC-1R cells than each single knockdown (Figure 7). This result suggested that the combination of MARCKS and WLS influenced the oxaliplatin resistance in pancreatic cancer. Based on quantitative proteomic analysis, resultingly, we demonstrate the possibility of multiple cross-talk signals and provide potential targets for the development of anti-chemoresistance drugs. Further studies are necessary for excavating other mechanisms in the regulation of the chemoresistance against anti-cancer drugs (e.g., 5-FU, gemcitabine, and so on).

## 4. Materials and Methods

### 4.1. Experimental Design and Statistical Rationale

To perform quantitative proteomic analysis, the human pancreatic cancer PANC-1 cells and oxaliplatin-resistant PANC-1 (PANC-1R) cells were metabolically labelled with the heavy amino acids (^13^C_6_-Arg and ^15^N_2_^13^C_6_-Lys) for SILAC-Heavy and their light counterparts (^12^C_6_-Arg and ^14^N_2_^12^C_6_-Lys) for SILAC-Light, respectively. SILAC-labelled PANC-1 (heavy) and PANC-1R (light) cells were used for proteomic analysis. The proteomic dataset was obtained from three biological replicates with two technical replicates using on-line 2D-nLC-MS/MS. A total of six datasets were obtained, each consisting of 12 MS raw data files. MS raw data were processed using MaxQuant search engine 1.6.1.0. To perform appropriate statistical analysis, we considered only proteins that were quantified at least three times in six datasets. Student’s t-test was performed using the Perseus software 1.5.8.5. *p*-values less than 0.05 were considered statistically significant. All data showed a normal distribution and linear correlation between replicates (see Result section). For a detailed description of MS data processing and statistical analysis, see the data analysis in the experimental procedures sections.

### 4.2. Establishment of an Oxaliplatin-Resistant Pancreatic Cancer Cell Line

The human pancreatic cancer cell line, PANC-1, was obtained from the Korean Cell Lines Bank (KCLB, Seoul, Korea) and cultured in Dulbecco’s Modified Eagle’s Medium (DMEM, Capricorn Scientific GmbH, Germany) with 100 units/mL penicillin, 100 µg/mL streptomycin, and 10% fetal bovine serum (FBS) at 37 °C with 5% CO_2_ in a humidified atmosphere. Anticancer-drug resistant PANC-1 cells were established by means of increasing concentrations of oxaliplatin, as previously described (16857785, 27910856, 23349823). Oxaliplatin (O9512) was purchased from Sigma-Aldrich (St.Louis, MO, USA). To establish a stable pancreatic cancer cell line chronically resistant to oxaliplatin, the PANC-1 cells were cultured at a starting concentration of 20 μg/mL oxaliplatin for 48 h. When the surviving population of PANC-1 cells became 80% confluent, the cells were sub-cultured twice. The concentration of oxaliplatin in the surviving PANC-1 cells was exposed to a stepwise increase in the same manner to 40 μg/mL, and finally to a concentration of 80 μg/mL. The surviving PANC-1 cells with final treatment of oxaliplatin were named PANC-1R. The sensitivity of parental PANC-1 and oxaliplatin-resistant PANC-1R cells to oxaliplatin was determined by cell viability assay analyzed by treatment for 48 h with different concentrations of oxaliplatin.

### 4.3. Cell Viability Assay

Cells were seeded in 96-well plates at a density of 1 × 10^4^ cells/well. Oxaliplatin was treated for 48 h at 37 °C with 5% CO_2_ in a humidified atmosphere. 10 μL/well of Ez-cytox (10 μL/well, Dogen bio, Seoul, South Korea) was incubated at 37 °C for 3 h. To measure the number of viable cells, the absorbance of each well was detected at 450 nm using an Epoch-2 microplate reader (BioTek, Winooski, VT, USA). The assays were performed in triplicate.

### 4.4. Colony Forming Assay

Equal numbers of PANC-1 or PANC-1R cells (1000/well) were seeded into 6-well plates and cultured for 2 weeks in the medium. After washing with phosphate-buffered saline (PBS), the cells were fixed with 4% paraformaldehyde for 30 min and stained with 0.1% crystal violet (C0775, Sigma-Aldrich) for 30 min at room temperature. The number of colonies was counted under a light microscope.

### 4.5. Stable Isotope Labelling with Amino Acids in Cell Culture (SILAC)

PANC-1cells were cultured in SILAC DMEM medium (Welgene, Daegu, Korea) with dialyzed FBS (Gibco, MA, USA) containing heavy 0.798 mM lysine and 0.398 mM arginine. Heavy lysine (1G: CLM-265-H-1) and arginine (1G: CNLM-291-H-1) were purchased from Cambridge Isotope Laboratories (CIL, Andover, MA, USA). PANC-1R cells were grown in light SILAC growth medium (DMEM, Capricorn Scientific GmbH, Germany) with dialyzed FBS (Gibco, MA, USA). All cells were maintained at 37 °C in humidified air containing 5% CO_2_. To validate labelling efficiency for full incorporation of heavy amino acid labels in all proteins, cells were cultured for seven passages and checked reached > 95% by LC-MS/MS analysis.

### 4.6. Sample Preparation for Proteomic Analysis

PANC-1 and PANC-1R cells were suspended with cell lysis buffer (8 M urea, 50 mM Tris-HCl pH 8.0, 75 mM NaCl, and a cocktail of protease inhibitors) and sonicated with ten 3-s pulses (2-s pause between pulses). The lysate was centrifuged for 15 min at 12,000 rpm, and the supernatant was collected for proteomic sample preparation. Protein concentrations were measured using a bicinchoninic acid (BCA) assay. An equal amount of proteins from PANC-1 and PANC-1R cells were mixed and followed by being reduced with 10 mM dithiothreitol (DTT) for 2 h at 37 °C and alkylated with 20 mM iodoacetamide (IAA) for 30 min at room temperature in the dark. The remaining IAA was quenched by the addition of excess L-cysteine. Samples were diluted with 50 mM ammonium bicarbonate buffer to a final concentration of 1 M urea and then digested with trypsin (1:50, w/w) for 18 h at 37 °C. To stop the digestion, 1% formic acid (FA) was added, and the resulting peptide mixtures were desalted with a 10 mg OASIS HLB cartridge (Waters, MA, USA). The eluted peptides were dried in a vacuum concentrator and reconstituted in 0.1% FA.

### 4.7. Liquid Chromatography-Tandem Mass Spectrometry (LC-MS/MS) Analysis

On-line 2D-nLC-MS/MS analysis was performed with a capillary LC system (Agilent Technologies, Waldbronn, Germany) coupled to a Q-Exactive™ Hybrid-Quadrupol-Orbitrap-mass spectrometer (Thermo Fisher Scientific, Bremen, Germany). For on-line 2D-nLC, biphasic reverse phase (RP)/strong cation exchange (SCX) trap columns were packed in one-end tapered capillary tubing (360 μm-O.D., 200 μm-I.D., 40 mm in length) with 5 mm of C18 resin (5 μm-200 Å) followed by 15 mm of SCX resin (5 μm-200 Å), as previously described [66]. The RP analytical column was packed in 150 mm capillary (360 μm-O.D., 75 μm-I.D.) with C18 resin (3 μm-100 Å). 

The peptides were injected into the trap column and fractionated with 12-step salt gradients (0, 15, 20, 22.5, 25, 27.5, 30, 40, 50, 100, 200, and 1000 mM ammonium bicarbonate buffer containing 0.1% FA. The stepwise elution of peptides from SCX resin were performed by each salt injection to trap column for 10 min. The eluted peptides were directly bound to the RP resin of the trap column and followed by 120 min RP gradients at a column flow rate 200 nL/min. The mobile phase consisted of buffer A (0.1% FA in water) and B (2% water and 0.1% FA in acetonitrile). The gradient was 2% B for 10 min, 2–10% B for 1 min, 10–17% B for 4 min, 17–33% B for 70 min, 33–90% B for 3 min, 90% B for 15 min, and 90–2% B for 2 min, and 2% B for 15 min.

The Q-Exactive mass spectrometer was operated in data-dependent mode. Full-scan MS spectra (*m/z* 300–1800) were acquired with automatic gain control (AGC) target value of 3E6 at a resolution of 70,000. MS/MS spectra were obtained at a resolution of 35,000. The top 12 most abundant ions from the MS scan were selected for high-energy collision dissociation (HCD) fragmentation with normalized collision energy (NCE) of 27%. Precursor ions with single and unassigned charge state were excluded. Dynamic exclusion was set to 30 s. Each biological replicate was analyzed in technical duplicate 2D LC runs.

### 4.8. Data Analysis

MS raw data files were analyzed with MaxQuant software (version 1.6.1.0,Max Planck Institute, Munich, Germany). MS peak lists were generated using MaxQuant. Generated MS peak lists were searched against the UniProt human database (3 January 2018 release) using the Andromeda search engine integrated in to the MaxQuant [67]. The search criteria were set as follows: Tryptic specificity was required; two mis-cleavages were allowed; the mass tolerance was 4.5 ppm and 20 ppm for precursor and fragment ions, respectively; carbamidomethylation of cysteine (C) was set as a fixed modification; oxidation of methionine (M) and acetylation of N-terminal residue was set as variable modifications; the false discovery rate (FDR) was set to 0.01 for both peptides and proteins; SILAC heavy label was set to Arg6 and Lys8. Only proteins were identified with at least two unique peptides per protein. All contaminants and reverse database hits were excluded from the protein list. Subsequent data processing and statistical analysis were performed using the Perseus software 1.5.8.5 [68]. The SILAC light/heavy ratios were log_2_ transformed and normalized by subtracting the median. To identify a significant difference between PANC-1 and PANC-1R cells, the Student’s t-test was applied. A functional Gene Ontology (GO) enrichment analysis was performed using DAVID. The enrichment analysis of the reactome pathway was performed using the R/Bioconductor package ReactomePA (version 1.30.0) [69]. A protein–protein interaction (PPI) network was constructed (medium confidence score, > 0.4) with the Search Tool for the Retrieval of Interacting Genes/Proteins (STRING) 11.0 and then visualized using Cytoscape software 3.7.1. Network module analysis was performed using the Molecular Complex Deletion (MCODE) plugin for Cytoscape. The parameters were set as degree cut-off = 2, node score cutoff = 0.2, k-core = 2, and maximum depth = 100.

### 4.9. Western Blot Analysis

Harvested cells were lysed in cold radioimmunoprecipitation assay (RIPA) lysis buffer [0.5 M Tris-HCl (pH 7.4) 1.5 M NaCl, 2.5% deoxycholic acid, 10% NP-40, 10 mM EDTA] with protease and phosphatase inhibitors (Gendepot, Katy, TX, USA). Cell lysates were analyzed by sodium dodecyl sulfate-polyacrylamide gel electrophoresis (SDS-PAGE) and transferred to nitrocellulose membranes (GE Healthcare, Milwaukee, Wisconsin, USA). After blocking with 8% skim milk or 5% bovine serum albumin (BSA) for 30 min, the membrane was probed with primary antibodies overnight at 4 °C. After washing with phosphate-buffered saline (PBS)/1% Tween-20 (T-PBS), the membrane was developed with a peroxidase-conjugated secondary antibody from Merck Millipore, and immunoreactive proteins were visualized using enhanced chemiluminescence reagents (Amersham Biosciences, Piscataway, NJ, USA), as recommended by the manufacturer. 

Primary antibodies were used for ISG15 (#2758), MARCKS (#5607), p-MARCKS (Ser159/163) (#11992), p-Akt (Ser473) (#9271), Akt (#9272), β-catenin (#9562), and cyclin D1 (#2978) from Cell Signaling Technology (CST, Beverly, MA). Primary antibodies were used for p53 (sc-126), IFIT3 (sc-393512), GAPDH (sc-47724), and HO-1 (sc-136960) from Santa Cruz Biotechnology (Santa Cruz, CA, USA). Anti-SOD2 (LF-PA0214) was obtained from Young In frontier (Seoul, Korea). Anti-β-actin (MAB1501) was obtained from Merck Millipore. Anti-WLS was obtained from Biolegend (San Diego, CA, USA).

### 4.10. RNA Isolation and qRT-PCR

Total RNA was purified using a TRIzol reagent (Ambion, Life Technologies, Carlsbad, CA, USA). 1ug of total RNA was synthesized to cDNA using a Prime Scrtipt^TM^ 1st strand cDNA synthesis (TaKaRa, Japan). The thermal conditions for reverse transcription PCR were as follows: Step 1, 65 °C for 5 min and 4 °C for 10 min; step 2, at 30 °C for 10 min, at 50 °C for 60 min, at 95 °C for 5 min, and 4 °C for 10 min. For analysis of relative quantitation, qRT-PCR reactions were subjected using TaKaRa SYBR Premix Ex Taq II (TaKaRa, Japan), and PCR processing was carried out in an iCycler (Bio-Rad, Hercules, CA, USA). The 10 μL of reaction contained 1 μL of cDNA, 5 μL of SYBR, 1.2 μL of primer mix, and 2.8 μL of water. The thermal conditions for qRT-PCR assay were as follows: 95 °C for 3 min; followed by 40 cycles of at 95 °C for 10 s, at 55 °C for 30 s, and at 95 °C for 10 s, at 65 °C for 0.05 s. The sequences of primers for human *MARCKS* were 5′-CCAGTTCTCCAAGACCGCAG-3′ (sense) and 5′-TCTCCTGTCCGT TCGCTTTG-3′ (antisense). The sequences of primers for human *WLS* were 5′-GCACCAAGA AGCTGTGCATT-3′ (sense) and 5′-GTTGTGGGCCCAATCAAGCC-3′ (antisense). The sequences of primers for *GAPDH* were 5′-TCGACAGTCAGCCGCATCTTCTTT-3′ (sense) and 5′-ACCAAATCCGTTGACTCCGACCTT-3′ (antisense). The copy number of these genes was normalized to an endogenous reference gene, *GAPDH*. The fold change from PANC-1 was set at 1-fold, and then the normalized fold change ratio was calculated. Data of relative gene expression was calculated by 2 ^ΔΔCT^ method [70].

### 4.11. siRNA Transfection

Cells were seeded in 6-well plates at a density of 4 × 10^5^ cells per well. After 16 h, the transfection was performed with 20 nM siRNA using 5 μL of Lipofectamine 2000 (Invitrogen, Carlsbad, CA, USA) in 3 mL of well per 6 well, according to the manufacturer’s protocol. The medium (3 mL per well of 6 well) was changed 8 h after transfection. After 48 h, transfected cells with the treatment of siRNA were harvested. siMARCKS (sc-35857), siWLS (sc-88713), and control siRNA (sc-37007) were purchased from Santa Cruz Biotechnology (Santa Cruz, CA, USA). Other siRNAs were designed and chemically synthesized by Genolution Pharmaceuticals (Seoul, South of Korea). The sequence of si MARCKS (#2 and #3) was as follows: Sense, 5′ UCAUUCAGGUCCAGAAACAUU 3′, antisense, 5′ UGUUUCUGGACCUGAAUGAUU 3′ and Sense, 5′ CUUCAAAGGACCCUAAACUUU 3′, antisense, 5′ AGUUUAGGGUCCUUUGAAGUU 3’. The sequence of si WLS (#2 and #3) was as follows: Sense, 5’ CUACAUGUCGGUGAAAUGUUU 3′, antisense, 5′ ACAUUUCACCGACAUGUAGUU 3′ and Sense, 5′ UGAAAUGGCCCAUGAAAGAUU 3’, antisense, 5′ UCUUUCAUGGGCCAUUUCAUU 3′. For viability against the treatment of oxaliplatin in transient knockdown of MARCKS or/and WLS, cells were seeded in 6-well plates at a density of 4 × 10^5^ cells per well. After 16 h, siRNAs were transfected at a final concentration of 20 nM with 5 μL of Lipofectamine 2000 for 48 h. Transient knockdown cells were trypsinized and planted in 96-well plates at a density of 1 × 10^4^ cells/well. After 16 h, oxaliplatin was treated for 48 h at 37 °C with 5% CO_2_ in a humidified atmosphere. 10 μL/well of Ez-cytox (10 μL/well, Dogen bio, Seoul, South Korea) was incubated at 37 °C for 3 h. To measure the number of viable cells, the absorbance of each well was detected at 450 nm using an Epoch-2 microplate reader (BioTek, Winooski, VT, USA). The assays were performed in triplicate.

### 4.12. Statistical Analysis

All experiments were conducted in triplicate, and the mean values ± standard deviation (SD) values were presented. Comparisons between the two groups were considered using the Student’s *t*-test. Differences between data groups were deemed statistically significant at *p* < 0.05.

## 5. Conclusions

The present study revealed the multifactorial mechanisms involved in oxaliplatin resistance in pancreatic cancer cells by performing a SILAC-based quantitative proteomic profiling. Moreover, functional studies demonstrated that up-regulation of MARCKS (Akt signaling) and WLS (Wnt/β-catenin signaling) contributes to the oxaliplatin resistance (Figure 8). Further investigation is required to elucidate detailed mechanisms, which will help to develop new therapeutic strategies for overcoming oxaliplatin resistance in the treatment of pancreatic cancer.

## Figures and Tables

**Figure 1 cancers-13-00724-f001:**
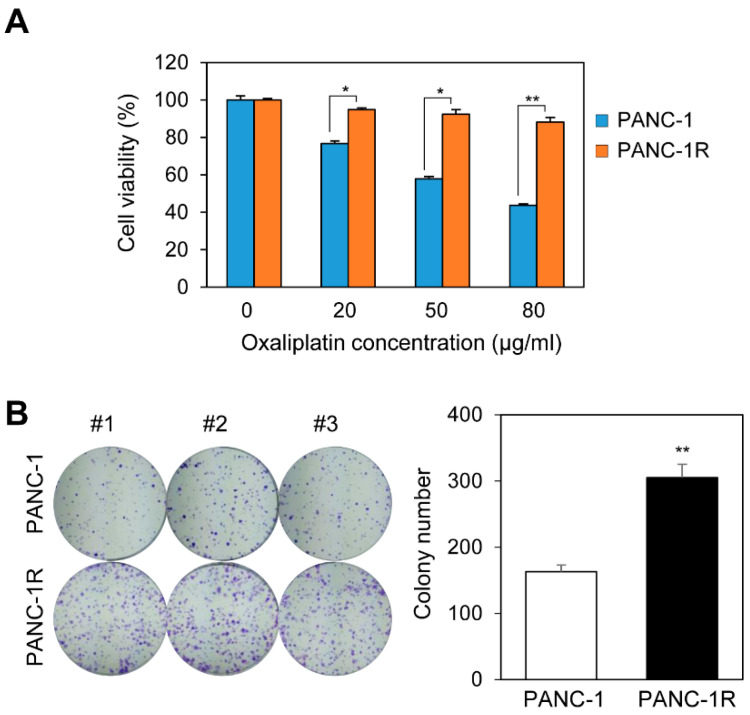
Establishment of oxaliplatin-resistant pancreatic cancer cell line. (**A**) Cellular viability was assayed by Ez-cytox on PANC-1 and PANC-1R with oxaliplatin for 2 days. (**B**) The colony formation assays were performed on PANC-1 and PANC-1R, respectively. * *p* < 0.05, ** *p* < 0.01.

**Figure 2 cancers-13-00724-f002:**
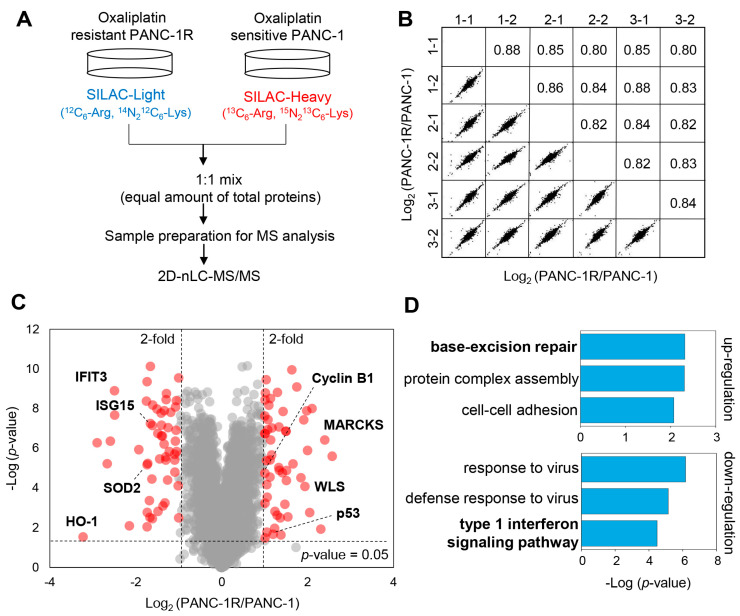
Proteomic comparison of oxaliplatin sensitive and resistant PANC-1 cells. (**A**) Proteomic workflow for SILAC labeling and LC-MS/MS. (**B**) Multiple scatter plots demonstrating reproducibility between the biological and technical replicates. Represented values are Pearson correlation coefficients. (**C**) Volcano plot showing the log_2_ fold-change and significance (−log_10_
*p*-value) of the proteome dataset. The cut-off values of fold-changes and significance is indicated with a dashed line. Red dots represent significantly changed proteins according to the *p*-value and fold-change cut-off values. (**D**) Top 3 significant Gene Ontology (GO) terms of up/downregulated proteins by biological process.

**Figure 3 cancers-13-00724-f003:**
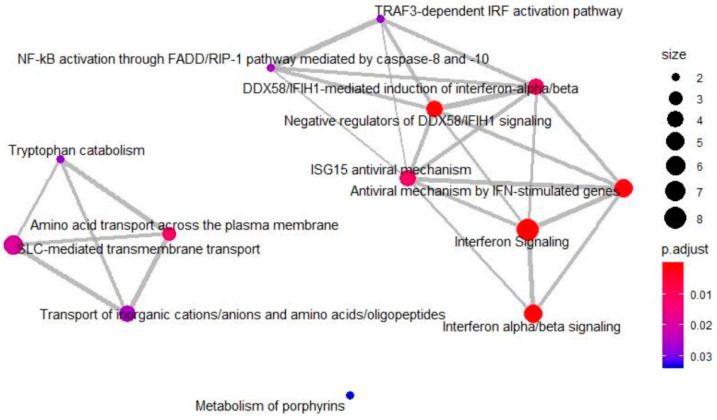
Reactome pathway enrichment map for downregulated proteins in oxaliplatin-resistant PANC-1 cells. The node color indicates significance of the reactome pathway and the node size represents the number of genes in the reactome pathway.

**Figure 4 cancers-13-00724-f004:**
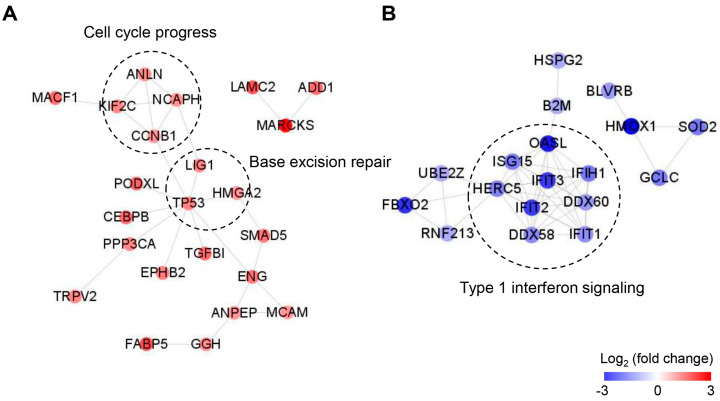
Protein–protein interaction (PPI) network analysis of up (**A**) and down (**B**) regulated proteins in oxaliplatin-resistant PANC-1 cells. The network was mapped using the STRING database and visualized by Cytoscape 3.7.2. Red nodes indicate up-regulation and blue nodes indicate down-regulation.

**Figure 5 cancers-13-00724-f005:**
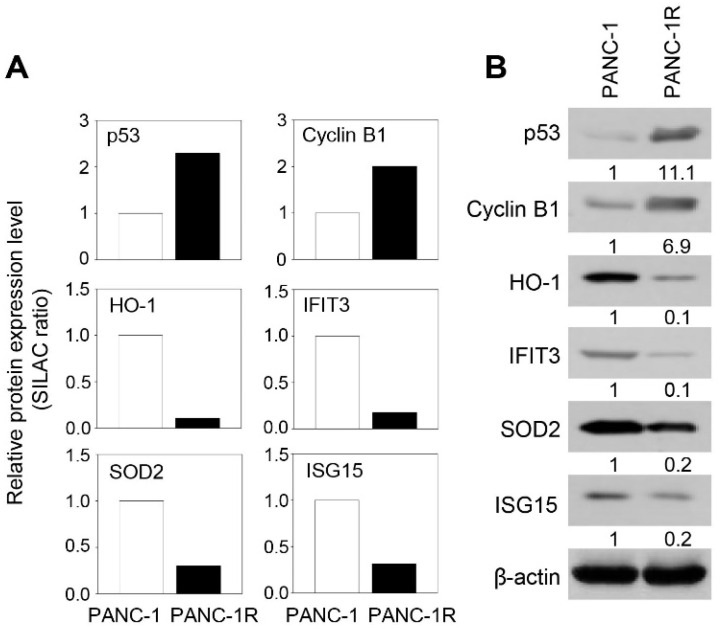
Validation of SILAC data by Western blot analysis. (**A**) Relative protein expression level of selected proteins (p53, cyclin B1, HO-1, IFIT3, SOD2, and ISG15) from SILAC data. Protein expression levels were normalized to oxaliplatin sensitive PANC-1 cells. (**B**) Validation of selected proteins (p53, cyclin B1, HO-1, IFIT3, SOD2, and ISG15) in both oxaliplatin sensitive and resistant PANC-1 cells by Western blot. β-actin was used as a loading control. Full-length Western blot images are presented in Appendix A.

**Figure 6 cancers-13-00724-f006:**
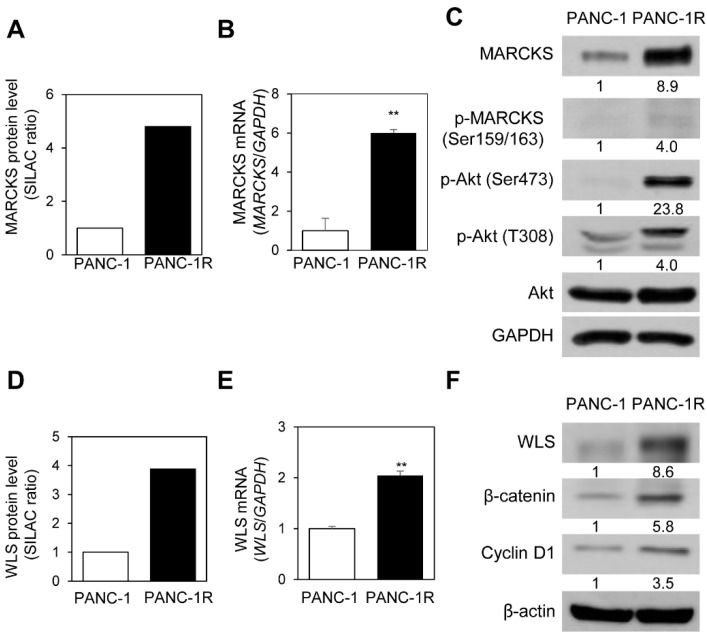
Myristoylated alanine-rich C-kinase substrate (MARCKS)- or Wntless homolog protein (WLS)-mediated downstream signaling is activated in PANC-1R cells. (**A**) The SILAC ratio of MARCKS was increased in PANC-1R cells. (**B**) The quantitative level of MARCKS mRNA by qRT-PCR was higher in PANC-1R cells. Three independent experiments were performed in triplicates. (**C**) The protein level of MARCKS, phospho-MARCKS (Ser159/163), phospho-Akt (Ser473 or Thr308), and total Akt was determined by Western blotting. GAPDH was used as a loading control. Full-length Western blot images are presented in Appendix A. (**D**) The SILAC ratio of WLS was increased in PANC-1R cells. (**E**) The level of WLS mRNA by qRT-PCR was higher in PANC-1R cells. Three independent experiments were performed in triplicate. (**F**) The protein level of WLS, β-catenin, and cyclin D1 was determined by Western blotting. β-actin was used as a loading control. ** *p* < 0.01. Full-length Western blot images are presented in Appendix A.

**Figure 7 cancers-13-00724-f007:**
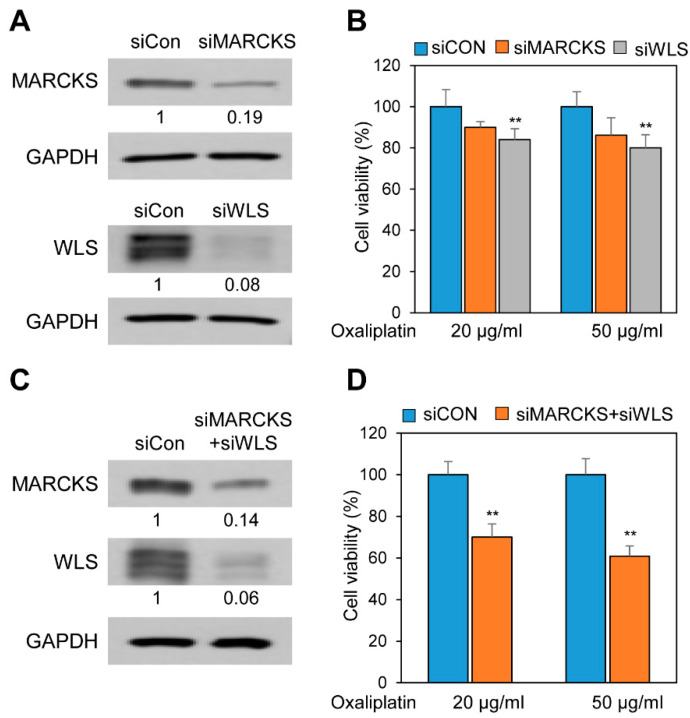
The inhibition of MARCKS and WLS induced PANC-1R cells to be sensitive to oxaliplatin. (**A**) The level of MARCKS or WLS in PANC-1R cell with the treatment of siMARCKS or siWLS was obtained by Western blotting. Full-length Western blot images are presented in Appendix A. (**B**) The cell viability to oxaliplatin was analyzed by Ex-cytox in PANC-1R with knockdown of MARCKS or WLS. (**C**) The levels of MARCKS and WLS in PANC-1R cells treated with both siMARCKS and siWLS were obtained by Western blotting. Full-length Western blot images are presented in Appendix A. (**D**) The cell viability to oxaliplatin was analyzed by Ex-cytox in PANC-1R cells treated with both siMARCKS and siWLS. ** *p* < 0.01.

**Figure 8 cancers-13-00724-f008:**
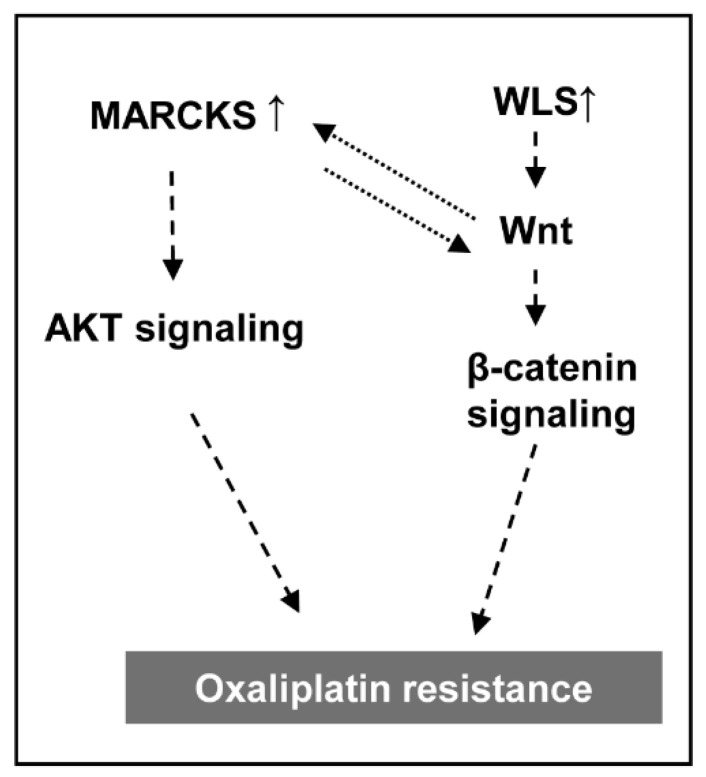
Schematic model of oxaliplatin resistance in pancreatic cancer cells. High expression of MARCKS induced activation of Akt signaling. High level of WLS induced β-catenin pathway by Wnt secretion. The crosstalk of MARCKS and WLS affected increased chemoresistance to oxaliplatin in the pancreatic cancer cell.

## Data Availability

The data presented in this study are available in Appendix A.

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
