# Peer review of "SILAC-Based Quantitative Proteomic Analysis of Oxaliplatin-Resistant Pancreatic Cancer Cells"

_cancers, 2021, doi:10.3390/cancers13040724_

Round 1

Reviewer 1 Report

The authors carefully addressed most of my comments and the manuscript got significant improved. I recommended its acceptance for publication. Just want to point out here that the supplementary figures are missing among the submission materials, please provide them accordingly.

Reviewer 2 Report

The authors have responded fully to the critiques and the paper is now acceptable for publication. Best regards,

Reviewer 3 Report

I reviewed the manuscript and agree it is ready for publication. Thanks

This manuscript is a resubmission of an earlier submission. The following is a list of the peer review reports and author responses from that submission.

Round 1

Reviewer 1 Report

In general, this is a very well written manuscript and the topic is certainly of interest to the readership of Cancers. In this report, the authors incorporated SILAC-based quantitative proteomics analysis to study the mechanism underlying oxaliplatin-resistance in pancreatic cancer. Two interesting targets MARCKS and WLS were identified to be upregulated in oxaliplatin-resistant PANC-1 cells, and their roles in controlling downstream signals and affecting sensitivity against oxaliplatin were subsequently studied.

While this report provided interesting insights in understanding oxaliplatin-resistance in pancreatic cancer, one major issue is that, there are some gaps between their data analysis (GO analysis, reactome pathway analysis and PPI network analysis) and target prioritization. For instance, what is the relation between the 6 proteins selected for Western validation with the pathway analysis? If purely based on changes in abundance, then what is the role of pathway analysis in guiding this study? That is the same case for MARCKS and WLS, if the author could provide more rationales in how they ended up selecting these two targets for detailed analysis based on pathway analysis, that would be helpful for the audience to understand the flow of this report.

In addition, I have several more detailed comments regarding to this manuscript, which are listed based on page/line number:

Page 1, line 28: the authors stated “we identified 107 proteins whose expression levels changed”, it would be more clear to specify the criteria here in selecting these 107 proteins (eg. 2 fold threshold, p-value ≤ 0.05).

Page 3, figure 1A: the asterisk marks for p-value seem to be mislabeled, for 20 and 50 ug/ml should be one*, 80 ug/ml should be two**.

Page 3, GO analysis: the GO analysis stated in this report seems to be very subjective/biased. For example, according to Table S4, in the subset of downregulated proteins, there are more targets fell in the “negative regulation of apoptotic process” category (with more counts and better p-value), however, the authors chose “positive regulation of apoptotic process” to report. Is there any rationale behind this selection? It would be helpful if the authors could provide more explanation here, not just list several GO terms.

Page 4, PPI network analysis: the same applied to PPI network analysis, the authors should provide more discussion about how this analysis would benefit the understanding of the mechanism behind oxaliplatin-resistance, not just simply list a few module/protein names.

Page 4, Figure 2A: the label for light and heavy Lys/Arg is not matching the ones described in main text/method, please correct the discrepancy.

Page 4, Figure 2C: it would be better to point out the key proteins selected for further studies in the figure, that would help the audience understand how the key proteins were prioritized.

Page 4, line 138-143: the content is duplicated for Figure 1 caption, please delete.

Page 6, verification: please discuss more about the rationale in choosing these 6 proteins, there is a big jump from the pathway/network analysis, seems like no clear correlation with the target prioritization.

Page 6, choose of MARCKS and WLS: same comment applied for MARCKS and WLS, the author need to provide more rationales in prioritizing these two key proteins.

Page 7, Figures 6B and 6D: please specify the reference gene (GAPDH) used in these assays in caption.

Page 7, siRNA knockdown: nowadays it is a general rule that at least 3 siRNAs were required to support knockdown assay (to exclude off-target effect). Please provide more evidence in supporting this key experiment. Also the authors stated “when silencing both MARCKS and WLS at the same time, the decrease in cell viability was significantly reduced compared to single gene silencing”, while I do see siWLS alone (Fig 7B) could already exert significant effect in reducing cell viability, maybe this statement is not quite accurate. In addition, as the double knockdown assay is key to this manuscript, it would be helpful if the author could provide stronger evidence in supporting the additive effect observed here, eg. tested with varying doses of oxaliplatin.

Page 8, Figure 8 and Page 13 conclusion: it is a quite weak and unclear conclusion here, maybe point out some potential directions based on the proteomics analysis. Also to support Figure 8, it would be helpful if the author could demonstrate knockdown MARCKS and/or WLS was able to shut off the downstream signals, that would support the direct effect of MARCKS and WLS upregulation.

Page 10, Establishment of an Oxaliplatin-Resistant Pancreatic Cancer Cell Line: is the cell identity confirmed by STR analysis? And whether the mycoplasma tested was regularly performed? In addition, is there any assay performed to confirm the resistant cell line was “stable” and “chronically resistant to oxaliplatin”?

Page 11, LC-MS/MS Analysis: please specify the duration for each salt gradient for SCX.

Page 12, RNA Isolation and qRT-PCR: please specify the conditions for cDNA synthesis and qPCR (temperature, duration, cycle number, etc.).

Page 13, siRNA transfection: please provide more detailed condition for siRNA transfection, eg. medium, ratio of siRNA/lipofectamine 2000. For assays determining the effect of siRNA knockout in sensitivity against oxaliplatin, please specify more details in how to set up the assay. Treat drug after 48 h transfection? Change media in between? Whether use 40 nM siCon as control?

Page 13, Supplementary Materials: please doublecheck the PRIDE accession number, the dataset can’t be found with PXD021251.

Author Response

Response to peer-reviewer

Followings are the original comments (Italic) with answers and corrections according to the reviewer's suggestions. Changes in the main text were highlighted in yellow.

Reviewer 1:

General notes

In general, this is a very well written manuscript and the topic is certainly of interest to the readership of Cancers. In this report, the authors incorporated SILAC-based quantitative proteomics analysis to study the mechanism underlying oxaliplatin-resistance in pancreatic cancer. Two interesting targets MARCKS and WLS were identified to be upregulated in oxaliplatin-resistant PANC-1 cells, and their roles in controlling downstream signals and affecting sensitivity against oxaliplatin were subsequently studied.

While this report provided interesting insights in understanding oxaliplatin-resistance in pancreatic cancer, one major issue is that, there are some gaps between their data analysis (GO analysis, reactome pathway analysis and PPI network analysis) and target prioritization. For instance, what is the relation between the 6 proteins selected for Western validation with the pathway analysis? If purely based on changes in abundance, then what is the role of pathway analysis in guiding this study? That is the same case for MARCKS and WLS, if the author could provide more rationales in how they ended up selecting these two targets for detailed analysis based on pathway analysis, that would be helpful for the audience to understand the flow of this report.

We profoundly appreciate both valuable comments and suggestions. We agree with the reviewer's comments concerning the gaps between proteomic data (GO, Reactome and PPI network analysis) and target prioritization. To this end, we revised the Figures (Figure 2D and 4), along with describing the rationale of selecting targets.

In GO analysis, we considered the top 3 ranked GO terms of biological process (Figure 2D) and focused on GO terms of “base-excision repair” and “type 1 interferon signaling pathway” in up/down-regulation. Also, we individually re-analyzed the PPI network of up and downregulated protein (Figure 4). Based on the PPI network analysis, we found that upregulated proteins are functionally implicated in cell cycle process. Hence, we took it into account those proteins that involve in base-excision repair, cell cycle process and type Ⅰ interferon signaling pathway in oxaliplatin resistance (in Results and Discussion sections). From this cause, we performed Western blot assays of p53 and Cyclin B1 (upregulated proteins in PANC-1R), which are functionally involved in base-excision repair and cell cycle process, along with Western blotting of downregulated proteins (IFIT3, ISG15, SOD2 and HO-1) implicated in type 1 interferon signaling pathway.

Out of up/downregulated proteins, this study aimed at figuring out other potential targets that could be one of factors in triggering oxaliplatin resistance in PANC-I cells, since it’s well-known that the up-regulation of base-excision repair and cell cycle progress contributes to oxaliplatin resistance in PANC-1 cells in previous studies. Out of top 10 most highly expressed proteins, we validated the functional roles of MARCKS and WLS in PANC-1R cells. MARCKS is a substrate of protein kinase C (PKC), and it has previously reported that inhibition of MARCKS overcome the drug resistance in multiple myeloma cells. In addition, the investigation of WLS –regulates the sorting and secretion of Wnt proteins – is also important, since the activation of Wnt signaling could be an underlying cause of drug resistance in cancer therapy.

Details of the revised figures and manuscript are described in following Comments 2,3,5,7 and 8 in Results section.

Abstract

Comment: (Page 1, line 28) the authors stated “we identified 107 proteins whose expression levels changed”, it would be more clear to specify the criteria here in selecting these 107 proteins (eg. 2 fold threshold, p-value ≤ 0.05).

As recommended, we described the threshold values of fold change and statistical significance to determine significantly changed proteins between PANC-1 and PANC-1R cells. (line 28, p1)

Results

Comment 1: (Page 3, figure 1A) the asterisk marks for p-value seem to be mislabeled, for 20 and 50 ug/ml should be one*, 80 ug/ml should be two**.

We corrected the asterisk marks for p-value (p3).

Comment 2: (Page 3, GO analysis) the GO analysis stated in this report seems to be very subjective/biased. For example, according to Table S4, in the subset of downregulated proteins, there are more targets fell in the “negative regulation of apoptotic process” category (with more counts and better p-value), however, the authors chose “positive regulation of apoptotic process” to report. Is there any rationale behind this selection? It would be helpful if the authors could provide more explanation here, not just list several GO terms.

We agree with reviewer’s comment. To improve the rationale of GO analysis, we only considered the top 3 ranked GO terms of biological process according to the statistical significance of the enrichment (Figure 2D). Among these, we focused on GO terms of base-excision repair and type 1 interferon signaling pathway in up/down-regulation and described the involvement of these GO terms in drug resistance. In addition, all statements about GO term of apoptosis signaling were removed in the revised manuscript.

  • Figure 2D and corresponding caption (p4): The top 3 ranked GO terms of biological process was shown in Figure 2D and corresponding caption was revised.
  • The sentence in the line 114, p3: The sentences were revised.
  • The sentence in the line 20, 30, 76 and 238: Multiple biological processes were exchanged to “DNA repair, cell cycle process and type Ⅰ interferon signaling pathway”

Comment 3: (Page 4, PPI network analysis) the same applied to PPI network analysis, the authors should provide more discussion about how this analysis would benefit the understanding of the mechanism behind oxaliplatin-resistance, not just simply list a few module/protein names.

We absolutely agree with reviewer’s comment. We re-analyzed the PPI networks of up and downregulated protein in PANC-1R cells, separately. In the PPI network of up-regulated proteins, the cluster (CCNB1-NCAPH-KIF2C-ANLN) functionally associated with mitotic cell cycle process was identified. In addition, we indicated base excision repair-related proteins in the PPI network. In the PPI network of down-regulated proteins, type Ⅰ interferon signaling proteins were clustered. In this section, we additionally described the involvement of cell cycle process in drug resistance.

  • Figure 4 and corresponding caption, p5: PPI networks of up and downregulated protein in PANC-1R cells were shown in Figure 4 and corresponding caption was revised.
  • The paragraph in the line 128, p4: The paragraph was revised.

Comment 4: (Page 4, Figure 2A) the label for light and heavy Lys/Arg is not matching the ones described in main text/method, please correct the discrepancy.

We apologize for the oversight. The SILAC label for light and heavy Lys/Arg has been corrected to “13C6-Arg/ 15N213C6-Lys for SILAC-Heavy and 12C6-Arg/ 14N212C6-Lys for SILAC -Light” in Figure 2A (p4).

Comment 5: (Page 4, Figure 2C) it would be better to point out the key proteins selected for further studies in the figure, that would help the audience understand how the key proteins were prioritized.

As suggested by the reviewer, the proteins selected for further studies were labelled with protein names on a volcano plot (Figure 2C, p4).

Comment 6: (Page 4, line 138-143) the content is duplicated for Figure 1 caption, please delete.

We apologize for the oversight. We have deleted the duplicate sentences for Figure 1 from Figure 2 caption (p4).

Comment 7: (Page 6, verification) please discuss more about the rationale in choosing these 6 proteins, there is a big jump from the pathway/network analysis, seems like no clear correlation with the target prioritization.

As the reviewer pointed out, we described the rationale in which proteins were selected for Western blot analysis. (line 161, p5)

Comment 8: (Page 6, choose of MARCKS and WLS) same comment applied for MARCKS and WLS, the author need to provide more rationales in prioritizing these two key proteins.

As the reviewer pointed out, we described the rationale in which MARCKS and WLS were selected for further study. (line 173, p6)

Comment 9: (Page 7, Figures 6B and 6D) please specify the reference gene (GAPDH) used in these assays in caption.

We corrected them according to your comments (p7).

Comment 10: (Page 7, siRNA knockdown) nowadays it is a general rule that at least 3 siRNAs were required to support knockdown assay (to exclude off-target effect). Please provide more evidence in supporting this key experiment.

We provided knockdown assays for two additional siRNAs against MARCKS or WLS. The sequence of siRNAs is described in the method section (p13), and the results are shown in Figure S3.

Also the authors stated “when silencing both MARCKS and WLS at the same time, the decrease in cell viability was significantly reduced compared to single gene silencing”, while I do see siWLS alone (Fig 7B) could already exert significant effect in reducing cell viability, maybe this statement is not quite accurate.

We have reflected on your comments and edited them (p7).

In addition, as the double knockdown assay is key to this manuscript, it would be helpful if the author could provide stronger evidence in supporting the additive effect observed here, eg. tested with varying doses of oxaliplatin.

We have reflected on your comments and edited them (p7-8). We tested cell viability of the double knockdown cell exposure to oxaliplatin (20 or 50 ug/ml). Cellular viability with 20 ug/ml of oxaliplatin was 30% lower in the double knock down group. The treatment of 50 ug/ml of oxaliplatin in the double knock down had 40% lower of viability compared to siCON. These results supported that the cross-talk between MARCKS and WLS affected chemoresistance to oxaliplatin.

Conclusions

Comment: (Page 8, Figure 8 and Page 13 conclusion) it is a quite weak and unclear conclusion here, maybe point out some potential directions based on the proteomics analysis. Also to support Figure 8, it would be helpful if the author could demonstrate knockdown MARCKS and/or WLS was able to shut off the downstream signals, that would support the direct effect of MARCKS and WLS upregulation.

We added states of the cross-talk between MARCKS and WLS and edited it in the discussion (line 295, p10). We also edited Figure 8 and more mentioned it in the figure legend of Figure 8 (p8).

Materials and Methods

Comment 1: (Page 10) Establishment of an Oxaliplatin-Resistant Pancreatic Cancer Cell Line is the cell identity confirmed by STR analysis? And whether the mycoplasma tested was regularly performed? In addition, is there any assay performed to confirm the resistant cell line was “stable” and “chronically resistant to oxaliplatin”?

Panc-1Ox cell line was authenticated by STR analysis and proved Panc-1 cell line (ATCC CRL-1469).

We tested the mycoplasma contamination using e-Myco™ Mycoplasma PCR Detection Kit (iNtRON, South of Korea) and did not detect it in cell lines.

Based on the establishment of the chemoresistance cell lines (Cancer Gene Ther 2016, 23, 446-453, PLoS One 2013, 8, e54193, Clin Cancer Res 2006, 12, 4147-4153), we constructed the chronically resistant to oxaliplatin using Panc-1 cell. The chemoresistant ability was usually validated by the calculation of IC50 increase and colony formation ability. We verified the chemoresistance of Panc-1Ox using the viability and colony formation exposure to oxaliplatin as in Figure 1. Resistant cell lines were verified through colony formation assay and cell viability assay for oxialiplatin that stably maintained resistance even after passage 10 times, and we did not use more than 10 passages.

Comment 2: (Page 11) LC-MS/MS Analysis: please specify the duration for each salt gradient for SCX.

As suggested by the reviewer, we described the detail information of SCX fractionation for online-2D-LC-MS/MS analysis (line 382, p12).

Comment 3: (Page 12) RNA Isolation and qRT-PCR: please specify the conditions for cDNA synthesis and qPCR (temperature, duration, cycle number, etc.).

We have reflected on your comments and explained conditions (p13).

Comment 4: (Page 13) siRNA transfection: please provide more detailed condition for siRNA transfection, eg. medium, ratio of siRNA/lipofectamine 2000.

We added the conditions according to your comments (p13).

For assays determining the effect of siRNA knockout in sensitivity against oxaliplatin, please specify more details in how to set up the assay. Treat drug after 48 h transfection? Change media in between? Whether use 40 nM siCon as control?

We have reflected on your comments and explained conditions (p13).

Supplementary Materials

Comment: (Page 13, Supplementary Materials) please doublecheck the PRIDE accession number, the dataset can’t be found with PXD021251.

We apologize for providing insufficient information regarding the PRIDE accession. The datasets are available in PRIDE, https://www.ebi.ac.uk/pride/archive/login. Reviewer account details are as follows:

Project accession: PXD021251

Username: reviewer36696@ebi.ac.uk, Password: kBkKiuuv

Reviewer 2 Report

The authors of this study generated an oxaliplatin-resistant cell population using Panc-1 cells and then performed SILAC-based proteomic study to evaluate differences between the cell lines. The proteomic practices and procedures seem reasonably well carried out and lead to clear conclusions with respect to a number of proteins up- and down-regulated in the derived, “resistant” cell line. Details of the methodology are provided. Appropriate tools are used and bioinformatic analysis carried out and displayed.

Questions/Concerns

Many questions and concerns are of the type one usually reaches in a study of this type. Does a single cell line represent a global response in pancreatic cancer? Are the changes observed comparable to acquired resistance in a patient? Does derivation of the resistant cell line really change the gene expression profile as a temporary response or a lasting response?  The study results themselves are well documented, and not every question is answerable by a single study.

Second, are the proteomic techniques themselves state-of-the-art? Is there adequate depth of discovery achieved that confer on the study a real step forward in discovering possible mechanisms of chemotherapeutic resistance? It would seem so. As the authors say, more studies are needed.

Lists of proteins detected, proteins elevated and decreased are provided. Schemes depict some of their potential physical/biologic interactions. Further validations are carried out on a subset of the findings. These provide a limited amount of understanding to the overall findings, for example upregulation of cyclin B helps understand the increased clonogenicity of the derived line. Finally, one or two elevated proteins were highlighted and their possible contribution to resistance substantiated by a limited rescue of the sensitive phenotype through knockdown approach. These strengths make the study more cohesive and "wrap up" although not in a totally satisfying way. Overall, we are inclined to support this type of exploratory work especially as it leaves ample clues for future consideration. Suggestion: the authors could evaluate the extent to which the elevated MARCKS is phosphorylated in assessing its contribution to Akt activation.

Minor detail. Part of the Fig. 2 legend appears to apply to Fig. 1. There are also misspellings in the tabs of the Supplemental data file. The writing is overall very good, but could be refined somewhat.

Author Response

Response to peer-reviewer

Followings are the original comments (Italic) with answers and corrections according to the reviewer's suggestions. Changes in the main text were highlighted in yellow.

Reviewer 2:

Comment 1: Many questions and concerns are of the type one usually reaches in a study of this type. Does a single cell line represent a global response in pancreatic cancer? Are the changes observed comparable to acquired resistance in a patient? Does derivation of the resistant cell line really change the gene expression profile as a temporary response or a lasting response?  The study results themselves are well documented, and not every question is answerable by a single study.

We absolutely agree with reviewer’s kind comments. Yes, the quantitative proteomic analysis of one type of cell line coupled with siRNA analysis did not fully represent all of the global responses in pancreatic cancer. However, this proteomics-based strategy can help and provide scientific clues to understanding either/both a specific or/and general mechanism for chemoresistance and other types of carcinomas.

In particular, Panc-1 cell used in this study is a primary tumor and also has low differentiation. Also, it’s common that metastatic pancreatic cancer cell lines (or other cancer cells) have been used to investigate chemoresistance mechanisms by means of grafting diverse anti-cancer drugs on a single cell line. Nowadays, we’ve planed diverse strategy to verify multi-mechanisms concerning chemoresistance that might be caused by other anti-cancer drugs such as gemcitabine, 5-FU, and irinotecan. Along with varying cell lines and anti-cancer drugs, therefore, this study on chemoresistance in pancreatic cancer cells be in need of further study, thereby resulting in the development of anti-chemoresistance therapies.

Viewed in similarity, chemoresistance of the acquired resistance in patients has the differenced expression of some factors such as RRM1, EGFR and so on (Clin Cancer Res. 2018 May 15;24(10):2241-2250). Also, ribonucleotide reductase catalytic subunit M1 (RRM1) is a potential marker of gemcitabine resistance. Erlotinib treatment in 3 trial revealed some patients had increased EGFR expression. In this study, we also detected the differential expression levels of these proteins. So, this indicated that a single cell-based proteomic approach might be alternative way to elucidate the mechanisms of chemoresistance.

Additionally, Changes in the levels of gene expression were a lasting response, not temporary. According to our experiments, the changed characteristics of the gene expression consistently maintained until over 20 passages of Panc-1Ox.

Comment 2: Second, are the proteomic techniques themselves state-of-the-art? Is there adequate depth of discovery achieved that confer on the study a real step forward in discovering possible mechanisms of chemotherapeutic resistance? It would seem so. As the authors say, more studies are needed.

In general, quantitative proteomic analysis of biological samples (e.g., cells, tissues, sera, and so on) has been considered to pave the way for profiling and figuring out key proteins that play a key role in biological mechanism(s) of interest. Unfortunately, it’s not possible that only one proteomic analysis of chemoresistant PANC-1R cells provides the complete information on chemotherapeutic resistance. From this cause, various methods (e.g, Western blotting, qRT-PCR and siRNA analysis, and so on) used in this study are inevitably required further verification tests to determine whether potential protein candidate is involved in the mechanism of interest or not. Also, as recommended by reviewer, there’s necessary more additional studies in order to provide the complete information on the reasons of occurring chemotherapeutic resistance. So, we revised in the last part of Discussion as “Further studies are necessary for excavation other mechanisms in the regulation of the chemoresistance against anti-cancer drugs (e.g., 5-FU, gemcitabine, and so on)

Comment 3: Lists of proteins detected, proteins elevated and decreased are provided. Schemes depict some of their potential physical/biologic interactions. Further validations are carried out on a subset of the findings. These provide a limited amount of understanding to the overall findings, for example upregulation of cyclin B helps understand the increased clonogenicity of the derived line. Finally, one or two elevated proteins were highlighted and their possible contribution to resistance substantiated by a limited rescue of the sensitive phenotype through knockdown approach. These strengths make the study more cohesive and "wrap up" although not in a totally satisfying way. Overall, we are inclined to support this type of exploratory work especially as it leaves ample clues for future consideration.

We have reflected on your comments in the Discussion section. We have added the role of upregulated p53 and cyclin B in the contribution to chemoresistance in the discussion (p9). We have added explanations of the necessary further studies in the discussion.

Comment 4: (Suggestion) the authors could evaluate the extent to which the elevated MARCKS is phosphorylated in assessing its contribution to Akt activation.

We detected phosphorylated MARCKS using western blotting assay and attached in Figure 6 of the manuscripts (p7).

Comment 5: (Minor detail) Part of the Fig. 2 legend appears to apply to Fig. 1. There are also misspellings in the tabs of the Supplemental data file. The writing is overall very good, but could be refined somewhat.

We apologize for the oversight. We have deleted the duplicate sentences for Figure 1 from Figure 2 caption (p4) and correct misspellings in the tabs of the Supplemental data file.
